# Smoothing Deep Reinforcement Learning for Power Control for Spectrum Sharing in Cognitive Radios

**Wanli Wang**

Department of Electronic and Information Engineering
The Hong Kong Polytechnic University
Hong Kong
`1155160517@link.cuhk.edu.hk`

## Abstract

The spectrum sharing in a cognitive radio system is related with a secondary user sharing common spectrum with a primary user for power transmit without inducing harmful inference. The deep reinforcement learning has been considered as an intelligent power control method via an agent continuously interacting with environment. Traditional deep $Q$-network in the frame work of deep reinforcement learning utilizes a deep neural network for learning a nonlinear function which maps the state or observation to accumulated rewards conditional on current state and agent action also called $Q$-value. The state or observation in the radio system is collected from wireless network and corrupted by noises. The deep neural network may therefore yield undesirable result due to the presence of noises and induced degraded network parameters. Considering that the kernel-based adaptive filter is beneficial for adaptive filtering, we aim to apply the kernel-based adaptive filter into traditional deep $Q$-network for smoothing network outputs. In addition, a weighting approach on the basis of past $Q$-values also works together with the deep neural network for further network output smoothing. The weighting approach is especially beneficial for alleviating the over-smoothing issue of the kernel-based adaptive filter. Simulation results have shown the efficiency of the proposed smoothing deep $Q$-network in spectrum sharing in cognitive radios in comparison with traditional deep $Q$-network.

**video link:**
**https://drive.google.com/file/d/1yomrCRflQ344ubQPY-eZYHcN4PosRuCt/view?usp=sharing**

## 1   Introduction

There is an urgent need to enhance the spectrum efficiency with increasing demand for spectrum resources or low utilization rate of some bands in a cognitive radio system. Consider a cognitive radio system consisting of a primary user and a secondary user [1 ,2]. A passive primary user model and an active primary user model are common operations of spectrum sharing [3]. For the passive primary model, the secondary user performs spectrum sensing to explore idle spectrum and there is no need for the primary user updating its transmission parameters. In the active model, however, dynamic power control strategies are required for all users such that a minimum quality of service (QoS) for successful data transmission is fulfilled for both primary and secondary users, which is considered in our work.

Apart from solving the power control issue from an optimization perspective [4 ,5], reinforcement learning [6 ,7, 8] as a subfield of machine learning [9] has gained popularity and been applied in intelligent power control due to its theoretical and technical achievements in efficiency and gener-

alization. The reinforcement learning problem is in general modeled as a Markov decision process (MDP) which is defined over the state transition probabilities and rewards dependent only on the state of environment and the action taken by the agent [10]. The reinforcement learning aims for behavioural decision making via interacting with the real world and receiving reward feedback. More concretely, the agent in reinforcement learning chooses an action on the basis of its current state and then receives feedback and reaches a new state. Therefore, the key point of reinforcement learning is about learning an optimal policy mapping the state to one agent action so that accumulated reward in the future is maximized.

Common reinforcement learning algorithms are categorized into model-based and model-free methods according to whether a complete knowledge of the MDP model is available [11, 12, 13]. Model-based methods, also referred to as planning methods, require a complete description of the model in terms of the transition and reward functions, while model-free methods, also referred to as learning methods, learn an optimal policy based on received observations and rewards. The model-based techniques mainly include two different approaches, i.e., value iteration and policy iteration [14]. However, the mathematic model is not always tractable in most applications. For solving this issue, model-free methods are developed for learning optimal policy via interacting with environment. Commonly used model-free approaches include Monte Carlo [11], temporal difference (TD) [11], SARSA [11, 15] and $Q$-learning [16, 17]. In particular, $Q$-learning is one of the most popular model-free approach updating $Q$-values in the form of a $Q$-table due to its simplicity and efficiency.

However, it is not feasible for using a look-up table for listing the expected $Q$-value for a pair of state and action with large number of states and actions. For dealing with this issue, the deep $Q$-network (DQN) under the framework of deep reinforcement learning replaces the $Q$-table by a deep neural network [18, 19]. In deep $Q$-network, the deep neural network is utilized for yielding target $Q$-value and estimated $Q$-value on the basis of agent states and received rewards which are stored at an experience pool [20]. The network parameters are then optimized via the nonlinear function of target $Q$-value and estimated $Q$-value [21].

The deep $Q$-network indeed behaves high efficiency in finding optimal policy but may suffer from noises in the process of updating network parameters, yielding degraded network outputs. This motivates us to combine deep $Q$-network with kernel-based adaptive filters [22]. Kernel methods aim to solve nonlinear filtering in a linear form in the reproduce kernel Hilbert space (RKHS) [23, 24, 25]. Commonly used kernel adaptive filters (KAFs) include the kernel least mean square algorithm (KLMS) [26], kernel affine projection algorithm (KAPA) [27] and kernel recursive least squares (KRLS) [28]. Among these algorithms, the KLMS performs desirable filtering performance with the lowest computational cost which is considered in our work. It is, however, not an easy work for the KLMS to choose an appropriate step size. A small step size may be not enough for the KLMS to behave desirable smoothing efficiency. By contrast, a large step size is indeed beneficial for smoothing but the KLMS may be trapped in over-smoothing issue. Therefore, a weighting approach on the basis of past $Q$-values is further considered. In the weighting approach, each sample within the experience pool is equipped with a sub-pool which stores its corresponding past $Q$-values. These past $Q$-values are considered for yielding a new smoothing output which is also beneficial for alleviating over-smoothing issue. A novel smoothing deep $Q$-network (SDQN) is therefore proposed by incorporating the KLMS algorithm and the weighting approach into deep $Q$-network for smoothing the outputs produced by the deep neural network. Simulation results have shown the efficiency of the proposed smoothing deep $Q$-network in the application of spectrum sharing in cognitive radios.

The remaining part of the paper proceeds as follows. Section 2 gives a brief overview of spectrum sharing in the cognitive radio system. Section 3 describes traditional deep reinforcement learning and its application in cognitive radios. Section 4 proposes a novel smoothing deep $Q$-network. Section 5 investigates the efficiency of the proposed smoothing deep $Q$-network in spectrum sharing in the cognitive radio system. Section 6 gives a conclusion.

## 2 Review of Spectrum Sharing in Cognitive Radios

Consider a cognitive radio network which is comprised of a primary user and a secondary user, respectively. Both users are expected to transmit their data successfully with required quality of service (Qos) [3]. In particular, the primary user adjusts its transmit power only on the basis of its own power control policy. In consideration of improving the efficiency of spectrum sharing, the

secondary user is expected to intelligently share a common spectrum resource with the primary user without inducing harmful inference to the primary user. Assume that there is no communication between the primary user and secondary user. This means that the secondary user has no knowledge about the power control policy of the primary user.

Consider the transmit power of the primary and secondary user by $p_1$ and $p_2$, respectively. The required quality of service (Qos) is measured from aspect of the signal-to-interference-plus-noise ratio (SINR) for the primary user or secondary user receivers which is defined by [29]

$$\text{SINR}_i = \frac{|h_{ii}|^2 p_i}{\sum\limits_{j \neq i} |h_{ji}|^2 p_j + \alpha_i}, i = 1, 2 \tag{1}$$

Notation $h_{ji}$ represents the channel gain from the transmitter $\text{Tx}_j$ to the receiver $\text{Rx}_i$ with noise power $\alpha_i$ at the receiver $\text{Rx}_i$. Both primary and secondary users are expected to transmit their data successfully with no less than the minimum SINR requirement for reception, i.e., $\text{SINR}_i \geq \eta_i, i = 1, 2$.

For the primary user, the power $p_1$ is adaptively adjusted by its own power control policy. Two common power control strategies are considered. In [20], the transmit power is updated as follows:

$$p_1(k+1) = D\left(\frac{\eta_1 p_1(k)}{\text{SINR}_1(k)}\right) \tag{2}$$

where $\text{SINR}_1(k)$ and $p_1(k)$ denote the SINR measured at the primary receiver and transmit power at the $k$-th time frame. The notation $D(\cdot)$ is used for discretizing continuous values into following discrete values

$$P_1 = \{p_1^p, \cdots, p_{L_1}^p\}, p_1^p \leq \cdots \leq p_{L_1}^p. \tag{3}$$

In addition, $D(x)$ is defined as the nearest discrete level no less than $x$. Assume the transmit power at the $k$-th time frame as $p_1(k) = p_j^p \in P_1$. The second strategy updates the transmit power by the following piece-wise function.

$$p_1(k+1) = \begin{cases} p_{j+1}^p & \text{if } p_j^p \leq \varpi \leq p_{j+1}^p \text{ and } j+1 \leq L_1 \\ p_{j-1}^p & \text{if } \varpi \leq p_{j-1}^p \text{ and } j-1 \geq 1 \\ p_j^p & \text{otherwise} \end{cases} \tag{4}$$

with $\varpi = \eta_1 p_1(k)/\text{SINR}_1(k)$.

The measurements which is related with the primary user and secondary user like the received signal strength (RSS) are collected from $N$ sensors in a wireless network. Denote the path loss between the primary transmitter, secondary transmitter and sensor $n$ by $g_{1n}$ and $g_{2n}$, formulated as

$$g_{mn} = \left(\frac{\zeta}{4\pi d_{mn}}\right), m = 1, 2 \tag{5}$$

with the signal wavelength $\zeta$ and $d_{1n}(d_{2n})$ representing the distance between the primary (secondary) transmitter and node $n$. Consider the following model for simulating RSS measurements.

$$P_n^r(k) = p_1(k)g_{1n} + p_2(k)g_{2n} + w_n(k), n = 1, \cdots, N \tag{6}$$

where $p_1(k)$ and $p_2(k)$ denote the transmit power of the primary and secondary user, respectively. Notation $w_n(k)$ represents a zero mean Gaussian noise with variance $\delta_n^2$ explaining for the random variation induced by shadowing effect and estimation errors. The secondary user takes the transmit power from the following finite set

$$P_2 = \{p_1^s, \cdots, p_{L_2}^s\}, p_1^s \leq \cdots \leq p_{L_2}^s. \tag{7}$$

An intelligent method is expected to let the secondary user adjust its own transmit power on the basis of available RSS information $\{P_n^r(k)\}_{n=1}^N$ so that both primary and secondary users meet their Qos requirements for power transmission.

# 3 Review of Deep Reinforcement Learning and Its Application in Cognitive Radios

Reinforcement learning is in fact described as a Markov decision process (MDP) $(\mathcal{S}, \mathcal{A}, \mathcal{P}, \mathcal{R}, \gamma)$ which consists of the agent's state $s \in \mathcal{S}$, action $a \in \mathcal{A}$, $\mathcal{P}$ describing the transition density $p(s^{'}|s, a)$ taking the action $a$ from the current $s$ to the next state $s^{'}$, $\mathcal{R}$ describing the instantaneous reward $r(s, a)$ and the discount factor $\gamma$. Since the mathematical model described by $\mathcal{P}$ is not always tractable in practice, a quadruple $(s, a, r, s^{'})$ in general characterizes one interaction between the agent and environment. More concretely, one policy $\pi$ provides a guideline for allowing the agent to choose an action given a state $s$. The agent then returns the reward $r$ as feedback from environment and updates the state from $s$ to $s^{'}$.

There is no doubt that the key of reinforcement learning is about the design of the optimal policy mapping the state $s$ to the expected action $a$. The expected long term reward given current state and action also called $Q$-value is expected to be maximized under the framework of value-based reinforcement learning. Consider the $Q$-value given state $s$, action $a$ and implicit policy $\pi$ starting from the discrete time $k$ [30].

$$Q_\pi(s, a) = E_\pi \left[ \sum_{t=0}^{\infty} \gamma^t r_{t+k+1} | s_k = s, a_k = a \right] \tag{8}$$

with the expectation operator $E(\cdot)$. Notation $r_{t+k+1}$ represents the instantaneous reward at the discrete time $t + k + 1$. The optimal $Q$-value is defined by $Q^*(s, a) = \max_\pi Q_\pi(s, a)$ and fulfills the following Bellman equation [31]

$$Q^*(s, a) = E \left[ r_{k+1} + \gamma \max_{a^{'}} Q^*(s_{k+1}, a^{'}) | s_k = s, a_k = a \right]. \tag{9}$$

Commonly used $Q$-learning updates $Q$-values in the form of the $Q$-table as follows:

$$Q(s, a) = Q(s, a) + \alpha \left( r + \gamma \max_{a^{'}} Q(s^{'}, a^{'}) - Q(s, a) \right). \tag{10}$$

However, it is not feasible for using a look-up table to list the expected $Q$-value for a pair of state and action with large number of states and actions. The loop-up table can be replaced by a deep neural network (DNN) parameterized by $\theta$, i.e., $Q(s, a; \theta)$, which is called deep $Q$-network (DQN) in the framework of deep reinforcement learning. In deep $Q$-network, an experience pool is constructed for storing a quadruples $(\mathcal{S}, \mathcal{A}, \mathcal{R}, \mathcal{S}')$ including current state $s \in \mathcal{S}$, taking the action $a \in \mathcal{A}$ from the current $s \in \mathcal{S}$ to the next state $s^{'} \in \mathcal{S}'$ and the instantaneous reward $r(s, a) \in \mathcal{R}$. The deep $Q$-network then aims for finding the following nonlinear function once a batch size of samples is randomly chosen from the experience pool.

$$f_\theta(s, a) = Q(s, a; \theta) \tag{11}$$

which maps the state into $Q$-values corresponding to multiple actions. In particular, the DQN optimizes the parameter $\theta$ in (11) by minimizing the difference between the output of the deep neural network and target, i.e., [20]

$$L_k(\theta) = \frac{1}{|\Omega_k|} \sum_{i \in \Omega_k} (Q_{ta}(i) - Q(s(i), a(i); \theta))^2 \tag{12}$$

where $\Omega_k$ and $|\Omega_k|$ denote the index set of the randomly chosen minibatch at the $k$-th iteration and its corresponding cardinality. In addition, the target $Q$-value $Q_{ta}(i)$ in correspond to $s(i)$ and $a(i)$ in (12) is constructed by

$$Q_{ta}(i) = r(i) + \gamma \max_{a^{'}} Q(s(i+1), a'; \theta), \forall i \in \Omega_k. \tag{13}$$

In (6), the RSS information $\{P_n^r(k)\}_{n=1}^N$ is collected from $N$ sensors and has a role of the state $s(k)$, i.e.,

$$s(k) = [P_1^r(k), P_2^r(k), \cdots, P_N^r(k)]. \tag{14}$$

As shown in (13), the deep $Q$-network aims to utilize the deep neural network for finding a well-learnt nonlinear function which maps the state $s(k)$ to $Q$-values corresponding to multiple agent actions. The action $a(k) = p_2(k+1)$ corresponding to the maximum $Q$-value is then taken for reaching the next state $s(k+1)$. The agent receives corresponding reward $r(k)$ as the feedback of taking the action $a(k) = p_2(k+1)$ where the reward $r(k)$ in the application of cognitive radios is set as

$$r(k) = \begin{cases} c, & \text{SINR}_1(k+1) \geq \eta_1 \text{ and } \text{SINR}_2(k+1) \geq \eta_2 \\ 0, & \text{otherwise} \end{cases} \tag{15}$$

with the positive number $c$ not too small and set as $c = 10$ in our work.

## 4  Smoothing Deep $Q$-Network

In deep $Q$-network, the target $Q$-value and estimated $Q$-value are all calculated by the deep neural network directly. The deep neural network may be not only sensitive to noises but also greatly affected by unsatisfactory trained network parameters, which yields undesirable $Q$-values. In consideration of adaptive filters beneficial for dealing with noises, it is natural to combine adaptive filters with the traditional deep neural network for producing smoothing $Q$-values. Since kernel adaptive filters exhibit superior filtering performance in comparison with adaptive filters, the kernel least mean square (KLMS) as one commonly used kernel adaptive filter is considered in our work due to its simplicity and desirable filtering precision.

There exist some issues to be addressed due to the incorporation of the KLMS. Although the KLMS exhibits superior filtering accuracy, the KLMS inherits the drawback of the kernel approach, i.e., linearly growing scale of the weight parameter. Different strategies can be adopted for alleviating or solving this issue like quantized kernel adaptive filters [32] or fixed budget quantized kernel adaptive filters [33]. In consideration of simplicity, the KLMS in this section adopts the way of discarding and re-training the parameter after samples are randomly chosen from the experience pool.

In addition, it is also necessary to notice that the KLMS is a commonly used supervised machine learning. This means that difference between target or label and its corresponding estimator is in general considered for updating the weight parameter of the KLMS. Therefore, it seems reasonable that the KLMS not only yields estimator but also constructs its own desirable target $Q$-value like those in deep $Q$-network. However, this may be not efficient. This is mainly because over-smoothing issue may occur if the KLMS yields estimator and target $Q$-value simultaneously. One way to alleviate this issue is about replacing the KLMS-based estimator or target value by the one yielded by the deep neural network. In our work, the former is considered which means only network-generated target $Q$-value is smoothed since the factor $\lambda$ in (10) is provided for scaling modified target $Q$-value. In consideration of alleviating computational burden, the KLMS adopts a single output for learning following nonlinear function:

$$f_{\theta_{klms}}(s') : s' \rightarrow \max_{a'} Q(s', a'; \theta) \tag{16}$$

where $\max\limits_{a'} Q(s', a'; \theta)$ represents the maximum $Q$-value produced by the deep neural network as shown in (13). As a result, the target $Q$-value in (13) is reformulated as

$$Q_{ta} = \rho \left( r + \gamma \max_{a'} Q(s', a'; \theta) \right) + (1 - \rho)$$
$$(r + \gamma Q_o(s'; \theta_{klms})) \tag{17}$$

where $Q_o(s'; \theta_{klms})$ represents the smoothing maximum $Q$-value produced by the KLMS.

As for aforementioned strategy, the network information is incorporated into the KLMS in the form of replacing the KLMS-based estimator by the network-generated estimated output and only the network-based target $Q$-value is smoothed. Inspired by this, a weighting approach by using past network-generated maximum $Q$-values is further considered for improving smoothing efficiency. As shown in (12), one experience pool is constructed by the deep neural network for storing a quadruple $(\mathcal{S}, \mathcal{R}, \mathcal{A}, \mathcal{S}')$ including the current state $s \in \mathcal{S}$, received reward $r \in \mathcal{R}$, adopted action $a \in \mathcal{A}$ and next state $s' \in \mathcal{S}'$. Apart from this experience pool, the sample with index $i, \forall i \in \Omega_k$ is

equipped with a sub-pool $Q^m(i)$ which stores its past maximum $Q$-values produced by the deep neural network. For example, a batch size of samples with the corresponding index set $\Omega_k$ is sampled from the experience pool for network training. The deep neural network therefore generates $|\Omega_k|$ maximum $Q$-values which are stored at corresponding $|\Omega_k|$ sub-pools, i.e., $Q^m(i), i \in \Omega_k$, respectively. These past maximum $Q$-values stored at corresponding sub-pools have a role of alleviating or avoiding the over-smoothing issue of the KLMS since these $Q$-values are all produced by the deep neural network. As a result, the smoothing $Q$-value for the sample with index $i, \forall i \in \Omega_k, Q_w(i)$ is calculated as follows:

$$Q_w(i) = \sum_{j=0}^{|Q_w(i)|-1} b^j Q_{m,|Q_m(i)|-j}(i) / \sum_{j=0}^{|Q_m(i)|-1} b^j \tag{18}$$

Notation $Q_m(i)$ and $|Q_m(i)|$ represent the sub-pool with the corresponding sample index $i, \forall i \in \Omega_k$ and its cardinality. Notation $Q_{m,|Q_i^m|-j}(i)$ denote the $|Q_m(i)| - j$-th element within the sub-pool with the corresponding sample index $i$. The parameter $b \in (0,1)$ is considered for scaling old $Q$-values at sub-pools. In consideration of not adding memory burden, it is necessary to discard old data for keeping the length of each sub-pool at $L$ when the corresponding length exceeds $L$.

The $Q$-target corresponding to the sample with index $i, \forall i \in \Omega_k$ is therefore re-formulated as

$$Q_{ta}(i) = \vartheta_e(i) \left( r + \gamma \max_{a'} Q(s(i+1), a'; \theta) \right) + \vartheta_s(i)(r+$$
$$\gamma Q_w(i)) + (1 - \vartheta_e(i) - \vartheta_s(i)) \left( r + \gamma Q_o(s(i+1); \theta_{klms}) \right) \tag{19}$$

with balanced parameters $\vartheta_e(i) \in (0,1)$ and $\vartheta_s(i) \leq 1 - \vartheta_e(i)$. As result, the KLMS in (19) is considered for smoothing the maximum $Q$-value yielded by the deep neural network. In addition, the weighting approach in (19) is combined with the KLMS for alleviating the issue of the over-smoothing issue. Apart from the weighting approach, additive noise is also beneficial for alleviating the over-smoothing by injecting uncertainty. In the following (20), additive noise is considered in spite of its limited efficiency in comparison with the weighting approach.

$$Q_{ta}(i) = \vartheta_e(i) \left( r + \gamma \max_{a'} Q(s(i+1), a'; \theta) \right) + \vartheta_s(i)(r+$$
$$\gamma Q_w(i)) + (1 - \vartheta_e(i) - \vartheta_s(i)) \left( r + \gamma Q_o(s(i+1); \theta_{klms}) \right)$$
$$+v(i) \tag{20}$$

where $v \sim N(\mu, \nu)$ represents the Gaussian noise with mean $\mu$ and variance $\nu$. Then, the network parameter $\theta$ is optimized by minimizing the loss function (12) by substituting $Q_{ta}(i)$ with (20).

The parameters $\vartheta_e(i)$ and $\vartheta_s(i)$ have a role of balancing the deep $Q$-network, KLMS and weighting terms and therefore essential for achieving satisfactory performance for nonlinear function learning. In particular, the parameter $\vartheta_e(i)$ is mainly used for balancing the deep $Q$-network term and smoothing one including the KLMS and weighting terms in (20). This motivates us to use the covariance function $\phi$ for designing the parameter $\vartheta_e(i)$. Consider an error variable $e = x - y$ with random variables $x$ and $y$. The nonlinear stationary covariance function $\phi$ takes the form of a squared covariance function, i.e., [34]

$$\ell_\beta^2(e) = \exp\left[ -\left( \frac{e}{\beta} \right)^2 \right] \tag{21}$$

where $\beta$ is the kernel size. As the squared covariance function in (21) is sensitive for the squared kernel width $\beta^2$ and squared error $e^2$, an exponential covariance function with kernel width $\beta$ can be considered, i.e., [34]

$$\ell_{\text{iso},\beta}(e) = \exp\left( -\frac{|e|}{\beta} \right) \tag{22}$$

which represents an *isotropic* covariance function with only one kernel width $\beta$ adopted for weighting the absolute value of $e$. In comparison with the isotropic covariance function, an *anisotropic* covariance function $\ell_{\text{ani},\beta}(e)$ is much more beneficial for dealing with the varying magnitude of the

absolute value of $e$, where the kernel width $\beta$ is controlled by a scale factor $\tau_{\text{iso}}$ and a secondary exponential covariance function, i.e.,

$$\ell_{\text{ani},\beta}(e) = \exp\left(-\frac{|e|}{\tau_{\text{iso}}\ell_{\text{iso},\beta}(e)}\right). \tag{23}$$

The parameters $\vartheta_e$ in (20) is therefore designed as

$$\vartheta_e(i) = \exp\left(-\frac{|e_{\vartheta_e(i)}|}{\tau_{\text{iso}}\ell_{\text{iso},\beta}(e_{\vartheta_e(i)})}\right). \tag{24}$$

The error variable $e_{\vartheta_e(i)}$ is defined by

$$\begin{aligned}
e_{\vartheta_e(i)} = &\max_{a'} Q(s(i+1), a'; \theta) - (zQ_o(s(i+1); \theta_{klms}) \\
&+ (1-z)Q_w(i))
\end{aligned} \tag{25}$$

with the balanced factor $z \in [0, 1]$.

However, the design of the parameter $\vartheta_e$ in (24) may be not always preferable for nonlinear function learning. Since the KLMS and weighting approach all focus on smoothing the outputs of the deep neural network, the outputs of the KLMS and weighting approach may be far away from that of the deep neural network, leading to $\vartheta_e \to 0$. This means that the deep $Q$-network term is neglected and smoothing term is highlighted as shown in (20). This is however not wise. It is uncertain that the KLMS and weighting approach perform better than the deep $Q$-network even if the smoothing output is far away from the network output. In fact, the smoothing term in (20) has no significant contribution on nonlinear function learning when the loss function changes slightly. The deep $Q$-network term therefore should be highlighted for alleviating over-smoothing. This motivates us to consider an switch scenario. In this scenario, the smoothing method should be dominant for smoothing $Q$-values when the loss function behaves great change. Otherwise, traditional deep $Q$-network is highlighted for prohibiting the algorithm to be trapped in over-smoothing. Consider the following ratio for explaining the change of the loss function.

$$\ell = \frac{|L_k(\theta) - L_{k-1}(\theta)|}{L_{k-1}(\theta)} \tag{26}$$

where $L_{k-1}(\theta)$ and $L_k(\theta)$ represent the loss function in (12) at the discrete time $k-1$ and $k$, respectively. When the loss function changes greatly, i.e., $\ell$ larger than the threshold parameter $\upsilon$, the parameters $\vartheta_e$ in (20) is designed as

$$\vartheta_e(i) = \exp\left(-\frac{|e_{\vartheta_e(i)}|}{\tau_{\beta_g}\ell_{\text{iso},\beta_g}(e_{\vartheta_e(i)})}\right) \text{ with } \ell > \upsilon \tag{27}$$

where $\ell_{\text{iso},\beta_g}(e_{\vartheta_e(i)})$ denotes the isotropic squared covariance function with kernel width $\beta_g$ and scaling factor $\tau_{\beta_g}$. Otherwise, the following parameter $\vartheta_e$ in (20) is considered when the loss function changes slightly, i.e., $\ell$ not larger than the threshold factor $\upsilon$.

$$\vartheta_e(i) = 1 - \exp\left(-\frac{|e_{\vartheta_e(i)}|}{\tau_{\beta_s}\ell_{\text{iso},\beta_s}(e_{\vartheta_e(i)})}\right) \text{ with } \ell \leq \upsilon \tag{28}$$

where $\ell_{\text{iso},\beta_s}(e_{\vartheta_e(i)})$ represents the isotropic squared covariance function with kernel width $\beta_s$ and scaling factor $\tau_{\beta_s}$.

In comparison with the parameter $\vartheta_e$, the parameter $\vartheta_s$ can be calculated in a much simpler manner. Since the parameter $\vartheta_s$ mainly focuses on balancing the KLMS and weighting approach which all focus on smoothing, they can be designed on the basis of deviating from the average smoothing output, i.e.,

$$\vartheta_s(i) = \exp\left(-\frac{|e_{\vartheta_s(i)}|}{\tau_{\beta_l}\ell_{\text{iso},\beta_l}(e_{\vartheta_s(i)})}\right) \tag{29}$$

with the scaling factor $\tau_{\beta_l}$. The error $e_{\vartheta_s(i)}$ is defined by

$$e_{\vartheta_s(i)} = (zQ_w(i) + (1-z)Q_o(s(i+1); \theta_{klms})) - Q_w(i) \tag{30}$$

where the balanced factor $z$ is as same as that in (25). Since the parameter $\vartheta_s$ is not larger than $1 - \vartheta_e$ as shown in (19), we have

$$\vartheta_s(i) = \min\left\{1 - \vartheta_e(i), \exp\left(-\frac{|e_{\vartheta_s(i)}|}{\tau\ell_{\text{iso},\beta_l}(e_{\vartheta_s(i)})}\right)\right\} \tag{31}$$

where the parameter $\vartheta_e$ is set according to (27) and (28).

**Algorithm 1** Smoothing Deep $Q$-network in Spectrum Sharing in Cognitive Radios

---

Initialize replay memory $D$ with buffer capacity $O$; network $Q(s; a; \theta)$ with random weights $\theta = \theta_0$.
Initialize the threshold parameter $\upsilon$ for determining whether (27) or (28) is considered.
Initialize $p_1(1)$ and $p_2(1)$, and generate $s(1)$.
**for** $k = 1, N$ **do**
   ▷ Update $p_1(k+1)$ by the primary user's power control strategy (2) or (4).
   ▷ Select an action $a(k)$ with the $\epsilon$-greedy strategy.
   ▷ Obtain $s(k+1)$ by the random observation model (6) and receive reward $r(k)$.
   ▷ Store transition $d(k) = \{s(k); a(k); r(k); s(k+1))\}$ in $D$.
   **if** $k \geq O$ **then**
      ▷ Sample a minibatch of transitions randomly $\{d(i)|i \in \Omega_k\}$ from $D$.
      ▷ Perform deep $Q$-network for calculating $\max_{a'} Q(s'(i+1), a'; \theta)$ in (13), $\forall i \in \Omega_k$.
      ▷ Store the maximum $Q$-value $\max_{a'} Q(s'(i+1), a'; \theta)$ at the sub-pool with the sample index
        $i$, i.e., $Q_m(i) = \{Q_m(i), \max_{a'} Q(s'(i+1), a'; \theta)\}, \forall i \in \Omega_k$.
      ▷ Perform the KLMS like (16) for computing $Q$-value $Q_o(s'; \theta_{klms})$ shown in (17) with
        the desired target $\max_{a'} Q(s'(i+1), a'; \theta)$ provided by traditional deep $Q$-network.
      ▷ Perform the weighting approach for the sample with index $i, \forall i \in \Omega_k$, and calculate
        $Q_w(i)$ in (18).
      ▷ Calculate the ratio $\ell$ in (26).
        **if** the ratio $\ell$ is larger than the threshold $\upsilon$ **then**
           Calculate the balanced parameter $\theta$ by (27).
        **else**
           Calculate the balanced parameter $\theta$ by (28).
        **end if**
      ▷ Calculate $Q$-target by (20).
      ▷ Update $\theta$ by minimizing the loss function (12) by substituting $Q_{ta}(i)$ with (20).
      ▷ Set $\theta_0 = \text{argmin}_\theta L(\theta)$.
   **end if**
   **if** $s(k)$ is a goal state **then**
      Initialize $p_1(k+1)$ and $p_2(k+1)$, and obtain $s(k+1)$.
   **end if**
**end for**

---

## 5 Simulation Result

In this section, we compare the performance of the proposed smoothing deep $Q$-network (SDQN) with that of traditional deep $Q$-network (DQN) in spectrum sharing in cognitive radios. The performance of the proposed method is evaluated via the success rate (SR) and corresponding ratio of success rate (RSR) between the DQN and SDQN. In particular, the success rate is calculated as the ratio of the number of successful experiments to the total number of runs. A successful experiment is considered as reaching a goal state within 20 iterations. The corresponding ratio of success rate (RSR) between the DQN and SDQN is defined by

$$\text{RSR} = \frac{\text{SR}_{\text{SDQN}} - \text{SR}_{\text{DQN}}}{\text{SR}_{\text{DQN}}} \tag{32}$$

where $\text{SR}_{\text{SDQN}}$ and $\text{SR}_{\text{DQN}}$ represent the success rates of the DQN and SDQN, respectively. In all experiments, we examine the success rate each 100 iteration for computational efficiency, i.e., $k_L = 100k$. Results are averaged over 50 independent runs, in which a random initial state is selected for each run.

Consider pre-designed sets $\mathcal{P}_1 = \mathcal{P}_2 = \{0.05, 0.1, 0.15, \cdots, 0.35, 0.4\}$ which provide transmit power for both primary and secondary users to choose. The channel gains from the primary user/secondary user transmitter to the primary user/secondary user receiver are set as $h_{ij} = 1, \forall i, j$. The noise power at the receiver $j, j = 1, 2$ is set to $\alpha_1 = \alpha_2 = 0.01$. For guaranteeing the successful reception for the primary user and secondary user, the minimum SINR requirements for the primary

user and the secondary are set as $\eta_1 = 1.2$ and $\eta_2 = 0.7$, respectively. In addition, $N$ sensors are considered for providing the secondary user with the RSS information for learning a power control policy. The distance $d_{ij}$ between the transmitter $\text{Tx}_i$ and the sensor node $j$ is set as uniformly distributed in the interval $(100, 300)$.

In all experiments, the deep neural network is configured as three hidden layers with the number of neurons 256, 256 and 512, respectively. Rectified linear units (ReLUs) are considered as the activation function for the first and the second hidden layers. The tanh function is used as the activation function for the last hidden layer. In our work, the total number of iterations is set to $K = 10^4$ and the experience pool contains $|D| = 320$ most recent transitions. The Adam algorithm is adopted for updating the weight $\theta$ with the minibatch size $|\Omega_k|$ in (12) set as 300 once the scale of the experience pool is larger than 300. The length of each sub-pool $L$ and scaling factor $b$ in (18) in the calculation of $Q_w(i)$ are set as $L = 15$ and $b = 0.95$, respectively. The probability of exploring a new action at the discrete time $k$ is set as $\epsilon_k = 0.8\,(1 - k/K)$ for the $\epsilon$-greedy strategy.

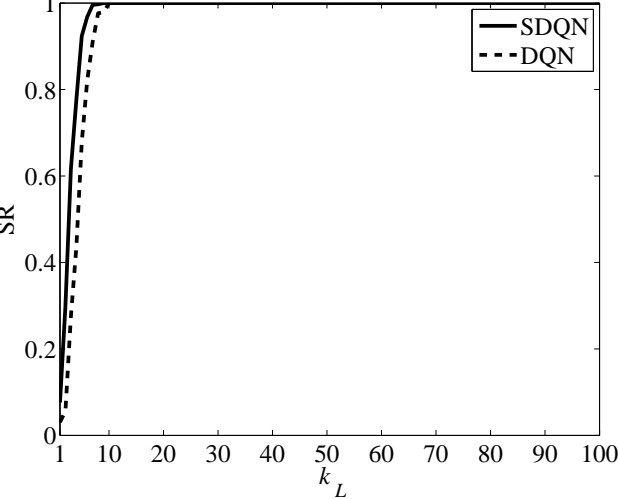

Figure 1: Success rate of DQN and SDQN with first policy for primary user updating its transmit power

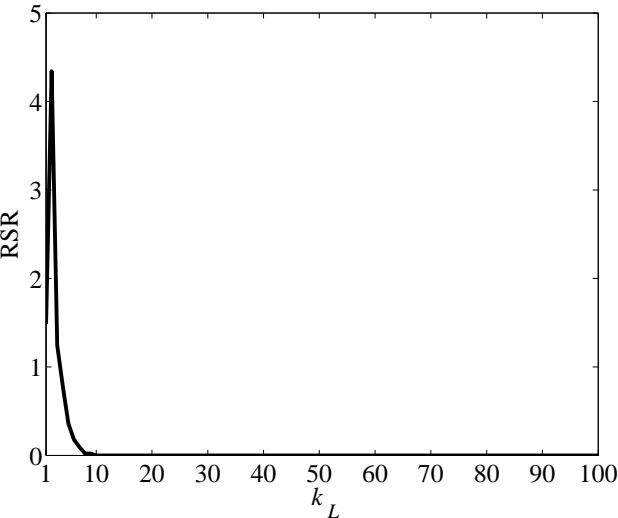

Figure 2: Ratio of success rate between DQN and SDQN with first policy for primary user updating its transmit power

Figs. 1–2 present the performance of the SDQN and corresponding RSR between the DQN and SDQN for learning a power control strategy when the primary user employs the first policy control

to update its transmit power in (6). In Figs. 1–2, the number of sensors $N$ is set as $N = 10$ and the standard deviation of noise $\delta_n$ in (6) is set as $\delta_n = \left(p_1^p g_{1n} + p_1^s g_{2n}\right)/10$. The anisotropic covariance functions in (27) and (28) are configured by kernel parameters $\beta_g = \beta_s = 10^4$ and scaling factors $\tau_{\beta_g} = 60$ and $\tau_{\beta_s} = 0.05$ for achieving a desirable balance between deep $Q$ network and smoothing terms including the KLMS and the weighting approach. The threshold parameter $\upsilon$ in (27) and (28) used for determining how to calculate $\vartheta_e(i)$ is set as $\upsilon = 0.2$. The kernel parameter $\beta_l$ and scaling factor $\tau_{\beta_l}$ in (29) are set as $\beta_l = 10^4$ and $\tau_{\beta_l} = 1$ for balancing the KLMS and weighting approach. In addition, the balanced parameter $z$ in (25) and (30) is all set as $z = 0.1$. The noise $v$ in (20) introduced for alleviating over-smoothing, is set as a Gaussian distributed noise with mean value $\mu = 0$ and variance $\nu = 0.3$. In Figs. 1–2, the proposed SDQN and traditional DQN all have the ability of reaching the highest success rate. However, the proposed SDQN achieves the highest success rate in a much faster rate than the DQN. In addition, the SDQN still has a higher success rate than the DQN before reaching the highest success rate like from $k_L = 1$ to $k_L = 10$.

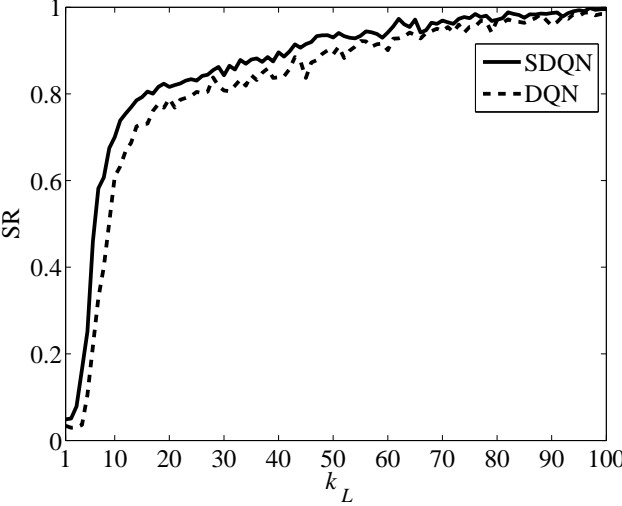

Figure 3: Success rate of DQN and SDQN with second policy for primary user updating its transmit power

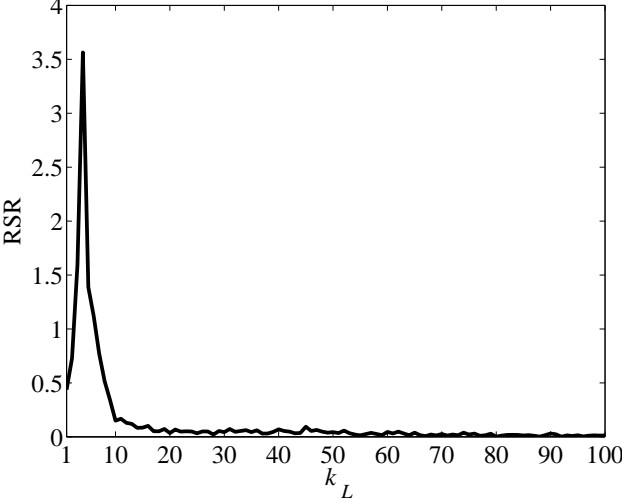

Figure 4: Ratio of success rate between DQN and SDQN with second policy for primary user updating its transmit power

In Figs. 3–4, the second policy control is considered for the primary user updating its transmit power like in (6). The settings of parameters are as same as those in Figs. 1–2. Figs. 3–4 show that the

proposed SDQN is much beneficial for the secondary user to learn an efficient power control policy when the primary user employs the second control policy for power transmission. Especially, the proposed SDQN has superior performance in comparison with the traditional DQN before reaching the highest success rate.

# 6  Conclusion

The spectrum sharing in cognitive radios is about the secondary user sharing common spectrum with the primary user without harmful inference to the primary user. The reinforcement learning has been an intelligent power applied in cognitive radios for spectrum sharing. The traditional deep $Q$-network in the framework of deep reinforcement learning may yield undesirable network outputs due to the presence of noises and degraded network weights. A novel deep $Q$-network called smoothing deep $Q$-network is therefore presented in this paper for improving the efficiency of the traditional deep $Q$-network. In the proposed smoothing deep $Q$-network, the kernel-based nonlinear filter is considered for smoothing the outputs of the deep neural network, beneficial for the agent taking optimal action in the process of interacting with environments. In addition, a weighting approach is also considered which applies past maximal $Q$-values to further smooth network outputs and also beneficial for alleviating the over-smoothing issue of the kernel-based adaptive filter. Simulation results have shown the efficiency of the proposed smoothing deep $Q$-network in the application of spectrum sharing in cognitive radios.

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
