# OpenReview forum: "Smoothing Deep Reinforcement Learning for Power Control for Spectrum Sharing in Cognitive Radios"
_CUHK.edu.hk/2021/Course/IERG5350_

### Official Review · AnonReviewer2 · 2020-12-19
**In this work a novel reinforcement learning algorithm named smoothing Deep Q-network is proposed and described clearly. Overall the work has high quality satisfying the requirements of the project and the work provides a good research direction.**

**Rating:** 8
**Confidence:** 3

**Review:**

In this work a novel reinforcement learning algorithm named smoothing Deep Q-network is proposed and described clearly. Overall the work has high quality satisfying the requirements of the project and the work provides a good research direction.

Pros:
	A novel reinforcement learning algorithm named smoothing Deep Q-network is proposed, the proposed algorithm is described clearly.
	The experiment and analysis are clear and easy to follow.
	The overall structure of the report is clear and well organized. The review of spectrum sharing in cognitive radios is described in detail.

Cons:
	Suggest you to cut the overall description about reinforcement learning in the introduction section and more focus on the introduction of your work. Since we all taken the course already, there is no need to give the overall description of reinforcement learning again.
	Suggest you to reduce the content of the review of reinforcement learning in section 3 and combine the section 2 and section 3 together to better describe how the problem is formulated as a reinforcement learning algorithm, what are the observation (state), action and reward, environment in your work, which helps the reader easier to follow.
	Suggest you to add some figures to better illustrate the spectrum sharing problem in section 2.
	Suggest you to read the paper “Intelligent power control for spectrum sharing in cognitive radios: a deep reinforcement learning approach”, which may help you to better describe.
	Suggest you to add a table to describe them in the appendix, which will help the readers to follow, as there are many symbols and variables in the paper.
	Suggest you to add subsections in the simulation result section, for example, baselines, evaluation metric, experimental settings, experiment results and analysis, which helps make content more clear.
	Suggest you to add description about the variable ρ in formula (17) on Page 5, immediately after you introduce the formula.
	Suggest you to rewrite “One way to alleviate this issue is about replacing the KLMS-based estimator or target value by the one yielded by the deep neural network.”in section 4 on Page 5, it is hard to understand the sentence for the first time.
	Suggest to add subsections in section 4, which will help make the description more clear and easy to follow.
	Small typo: “In addition, it is also necessary to notice that the KLMS is a commonly used supervised machine learning.”in section 4 on Page 5 (perhaps need to add method at the end).